# Thyroid Dysfunction under Amiodarone in Patients with and without Congenital Heart Disease: Results of a Nationwide Analysis

**DOI:** 10.3390/jcm11072027

**Published:** 2022-04-05

**Authors:** Alicia Jeanette Fischer, Dominic Enders, Lars Eckardt, Julia Köbe, Kristina Wasmer, Günter Breithardt, Fernando De Torres Alba, Gerrit Kaleschke, Helmut Baumgartner, Gerhard-Paul Diller

**Affiliations:** 1Department of Cardiology III—Adult Congenital and Valvular Heart Disease, University Hospital Muenster, D-48149 Muenster, Germany; g.breithardt@uni-muenster.de (G.B.); fernando.detorresalba@ukmuenster.de (F.D.T.A.); gerrit.kaleschke@ukmuenster.de (G.K.); helmut.baumgartner@ukmuenster.de (H.B.); gerhard.diller@ukmuenster.de (G.-P.D.); 2Institute of Biostatistics and Clinical Research, University Hospital Muenster, D-48149 Muenster, Germany; dominic.enders@ukmuenster.de; 3Department of Cardiology II—Electrophysiology, University Hospital Muenster, D-48149 Muenster, Germany; lars.eckardt@ukmuenster.de (L.E.); julia.koebe@ukmuenster.de (J.K.); k.wasmer@alexianer.de (K.W.)

**Keywords:** adult congenital heart disease, congenital heart disease, thyroid dysfunction, antiarrhythmic therapy, amiodarone

## Abstract

Background: Amiodarone has a profound adverse toxicity profile. Large population-based analyses quantifying the risk of thyroid dysfunction (TD) in adults with and without congenital heart disease (ACHD) are lacking. Methods: All adults registered with a major German health insurer (≈9.2 million members) with amiodarone prescriptions were analyzed. Occurrence of amiodarone-associated TD was assessed. Results: Overall, 48,891 non-ACHD (37% female; median 73 years) and 886 ACHD (34% female; median 66 years) received amiodarone. Over 184,787 patient-years, 10,875 cases of TD occurred. The 10-year risk for TD was 38% in non-ACHD (35% ACHD). Within ACHD, compared to amiodarone-naïve patients, the hazard ratio (HR) for TD was 3.9 at 4 years after any amiodarone exposure. TD was associated with female gender (HR 1.42, *p* < 0.001) and younger age (HR 0.97 per 10 years, *p* = 0.009). Patients with congenital heart disease were not at increased risk (HR 0.98, *p* = 0.80). Diagnosis of complex congenital heart disease, however, was a predictor for TD (HR 1.56, *p* = 0.02). Amiodarone was continued in 47% of non-ACHD (38% ACHD), and 2.3% of non-ACHD (3.5% ACHD) underwent thyroid surgery/radiotherapy. Conclusions: Amiodarone-associated TD is common and comparable in non-ACHD and ACHD. While female gender and younger age are predictors for TD, congenital heart disease is not necessarily associated with an elevated risk.

## 1. Introduction

Cardiac arrhythmias are frequent in the aging population with cardiovascular conditions. They are considered one of the most common complications in adult patients with congenital heart disease (ACHD) [1,2]. Beyond catheter ablation, pharmacological treatment with antiarrhythmic drugs represents the mainstay of therapy. Unfortunately, most Class I antiarrhythmic drugs seem contraindicated in patients with structural heart disease or ACHD or appear to be ineffective, such as beta-blockers or sotalol, thus limiting the pharmacological options in the majority of patients to amiodarone [3,4]. Although amiodarone has a high antiarrhythmic potency, it is associated with an unfavorable toxicity profile, specifically thyroid dysfunction [5,6]. Thyroid dysfunction in itself is associated with adverse outcomes, including a higher mortality in ACHD [7].

The mechanisms leading to thyroid dysfunction on amiodarone are multifactorial. Amiodarone is iodine-rich. Structurally, the drug resembles intrinsic thyroid hormones [8]. Amiodarone may cause immunologic injury and is associated with the formation of free radicals that further impair thyroid function [9]. Amiodarone decreases serum T3 (triiodothyronine) by blocking peripheral T4 (thyroxine) 5′-deiodinase [10]. This may result in increased T4-levels with low levels of T3 and a suppressed TSH (thyroid stimulation hormone), which may be misinterpreted as thyroid dysfunction.

Previous studies have suggested that ACHD patients as a cohort with a high arrhythmia burden and few medical alternatives to amiodarone may have a particularly high risk of amiodarone-induced thyroid disease [11,12], but this group of patients has not been directly compared to patients without ACHD. In ACHD patients, treatment of arrhythmic complications is of particular importance because of their marked impact on morbidity and mortality and the relative paucity of alternative drugs [1,13,14].

Using a large, nationwide administrative dataset, we aimed to clarify the frequency of thyroid complications under amiodarone treatment and to compare the occurrence between patients with and without ACHD. Furthermore, we aimed to identify independent risk factors for thyroid dysfunction after amiodarone intake and to assess therapeutic consequences of diagnosed amiodarone-induced thyroid dysfunction.

## 2. Materials and Methods

Data were acquired retrospectively from the administrative reimbursement database of the BARMER health insurance company that comprises ≈9.2 million insurance holders (out of ≈83 million German inhabitants). All cardiac and extracardiac diagnoses are encoded according to the German Modification of the International Statistical Classification of Diseases and Related Health Problems, 10th Revision (ICD-10-GM). Procedures are documented using the German Procedure Classification (OPS) codes. Drug prescriptions are encoded based on Anatomical Therapeutic Chemical (ATC) codes. Complete documentation is mandatory for reimbursement purposes and, thus, the analyzed dataset is complete for all insured individuals. Patient characteristics and co-morbidities were assessed based on in- and outpatient diagnostic and procedural codes (for a full list of ICD-10-GM, OPS-procedure and ATC drug codes used in this analysis, see Appendix A).

All patients ≥16 years old who received at least one prescription of amiodarone within the study period from 2005 to 2018 were included in the analysis. First prescription of amiodarone was considered the index-event for the study. For inclusion, data covering at least one year before the first prescription had to be present. Only patients without prior documentation of thyroid dysfunction, prescription of thyroid medication or amiodarone were included. Patients who left the insurance company within the study period were censored. Included patients were assigned to an ACHD or non-ACHD group based on the corresponding ICD-10-GM codes. Severity of congenital heart disease was graded as mild, moderate or severe according to current guidelines [15] (see Appendix A). Following a 10-day initiation period after first prescription of amiodarone to allow blood saturation of the drug, included patient datasets were screened for coding of thyroid dysfunction (defined as hypo-, hyperthyroidism, thyroiditis and/or thyroid surgery/radiotherapy) as well as de novo documentation of hypothyroidism or hyperthyroidism only (see study design Appendix A). Coding of other diagnoses of thyroid dysfunction such as subclinical hypothyroidism through iodine deficiency were not counted as thyroid dysfunction related to amiodarone intake.

### 2.1. Data Availability Statement

The data utilized in this study cannot be unblinded and shared in the manuscript, the Appendix A or in a public repository due to German data protection laws (‘Bundesdatenschutzgesetz’, BDSG). They are stored on a secure drive in the Barmer health insurance company database to facilitate replication of the results. Generally, access to data of statutory health insurance funds for research purposes is possible only under predefined conditions of the German Social Law (SGB V § 287). Requests for data access can be sent as a formal proposal specifying the recipient and purpose of data transfer to the appropriate data protection agency. Access to the data used in this study can only be provided to external parties under the conditions of the cooperation contract of this research project and after written approval by the health insurance company.

### 2.2. Statistical Analysis

Categorical variables are presented as numbers and percentages, while continuous variables are shown as median and interquartile ranges (IQR). Differences between groups were assessed by Mann–Whitney U-test or Fisher’s exact test depending on the data type.

For the endpoints of thyroid dysfunction, hypothyroidism and hyperthyroidism, only first events were considered. To assess risk factors of thyroid dysfunction, patient demographics, underlying condition, complexity of congenital heart disease as well as non-cardiac comorbidities and ongoing treatment with amiodarone (defined as prescription of amiodarone within the past 90 days) were studied using multivariable time-dependent Cox analysis. For the time-to-event analysis, multivariable Cox regression models that adjusted for patient characteristics at the index event of initial prescription of amiodarone were fitted. The models further included the time-dependent covariable ‘amiodarone intake’, defined as prescription of amiodarone within the past 90 days, to investigate the effect of active amiodarone therapy. Since the effect of amiodarone intake was increasing over time and thus violating the proportional hazards (PH) assumption, we allowed the corresponding coefficient to be time-varying by using a step function on 4 time intervals ([index, 1 year), [1 year, 2 years), [2 years, 3 years), [3 years, ∞)), which resolved the problem of non-proportional hazards. Results are summarized as hazard ratios (HR) with 95% confidence intervals (CI).

Furthermore, to assess the specific risk of thyroid dysfunction in ACHD patients, those after intake of amiodarone (current or previous) were compared to a matched amiodarone-naïve patient cohort using a propensity score-matching algorithm. A propensity score-matched control group of amiodarone-naïve ACHD patients was identified via a 1:1 matching. Propensity scores were calculated using logistic regression based on age, gender, complexity of congenital heart disease and different comorbidities, including the common risk factors defining overall morbidity of the respective patients. The matching procedure was successful for 782 out of 886 (88%) ACHD patients with amiodarone treatment since the propensity score distributions did not fully overlap.

All analyses were explorative, and a two-sided *p*-value < 0.05 was considered significant throughout the study. All analyses were performed using the statistical software R version 4.0.3 (R Foundation, Vienna, Austria).

## 3. Results

Overall, 101,570 patients ≥16 years old were treated with amiodarone throughout the study period (see Appendix A). Of those, 48,891 non-ACHD (37.0% female, median age 73.4 years [IQR 66.1–79.4]) and 886 ACHD patients (34.0% female; median age 65.9 years [IQR 55.0–74.7]) were included in this analysis. Baseline characteristics and demographic data of the analyzed individuals stratified by non-ACHD and ACHD are presented in Table 1.

ACHD patients were of a younger age (*p* < 0.001) and more frequently had right-sided heart failure (*p* = 0.01). There were no relevant differences in prevalence of left heart failure (*p* = 0.36). In non-ACHD patients, heart failure medication, with the exception of beta blockers, was prescribed more frequently compared to ACHD patients. More than one arrhythmia was documented in many individuals receiving amiodarone, reflecting the overall morbidity and arrhythmia burden of the analyzed patients.

Over a total of 184,787 patient-years (median observation time 2.7 years [IQR 1.0–5.7 years] in non-ACHD and 3.5 years [IQR 1.3–6.3 years] in ACHD patients), 10,677 events (21.8%) of thyroid dysfunction occurred in non-ACHD and 198 (22.3%) in ACHD individuals (*p* = 0.71, Table 2). The 10-year risk of any thyroid dysfunction was 38% in non-ACHD and 35% in ACHD patients (see Figure 1).

Irrespective of diagnosis of congenital heart disease, hypothyroidism was documented more frequently (14.5% non-ACHD versus 15.6% ACHD patients; *p* = 0.36) than hyperthyroidism (10.4% non-ACHD versus 11.6% ACHD patients; *p* = 0.24) in both cohorts.

Overall, the risk of thyroid dysfunction increased gradually with the duration of amiodarone intake in non-ACHD patients (HR 3.55 at four years of therapy, *p* < 0.001). In these, younger patient age (HR 0.97/10 years, *p* = 0.009), female gender (HR 1.42, *p* < 0.001), pacemaker therapy (HR 1.10, *p* = 0.006), implantation of implantable cardioverter defibrillators (HR 1.23, *p* < 0.001), chronic kidney disease (HR 1.49, *p* < 0.001), alcohol abuse (HR 1.17, *p* < 0.001) and smoking (HR 1.13, *p* < 0.001) emerged as significant risk factors for thyroid dysfunction on multivariable analysis. The presence of congenital heart disease per se was not significantly related to the risk for thyroid disease (HR 0.98, *p* = 0.80) (see Figure 2).

In the subgroup of ACHD patients, amiodarone intake was associated with an increased risk for thyroid dysfunction dependent on the duration of intake (HR 3.61 after 4 years of therapy, *p* < 0.001). Younger patient age (HR 0.87/10 years of age, *p* = 0.005) as well as female gender (HR 1.63, *p* = 0.001) emerged as risk factors for thyroid dysfunction in this subpopulation. Patients with severe complexity of congenital heart defect were at a significantly higher risk for thyroid dysfunction than patients with simple defects (HR 1.56, *p* = 0.02) (see Figure 3).

Separate analyses for hyper- and hypothyroidism for the overall and the ACHD cohort are shown in Appendix A. These confirmed the general associations of the overall analyses.

Compared to a matched amiodarone-naïve ACHD cohort, individuals with any amiodarone exposure had a significantly higher risk of acquiring thyroid dysfunction (HR 3.7 at 1 year, HR 3.9 at 4 years, *p* < 0.001) (see Table 3).

After establishing the diagnosis of thyroid dysfunction, catheter ablation was initiated in 12.7% of non-ACHD (n = 1352) and 15.7% of ACHD (n = 31) individuals. The number of patients receiving Class Ic antiarrhythmic drugs increased slightly from 1.9% to 2.1% in non-ACHD and 3.5% to 4.0% in ACHD patients after diagnosis of thyroid disease (Appendix A).

Of note, after diagnosis of thyroid dysfunction, amiodarone prescriptions were continued in 47.0% in non-ACHD and 37.9% in ACHD patients, with no relevant differences between the two groups (*p* = 0.08). Apart from hormone substitution or thyrostatic drug therapy, radiotherapy was initiated in 0.7% in non-ACHD and 1.5% in ACHD patients (*p* = 0.17) and thyroid surgery in 1.6% in non-ACHD and 2.0% in ACHD patients (*p* = 0.56) (see Table 4).

## 4. Discussion

Based on non-selective administrative data, the current study demonstrates that amiodarone-associated thyroid dysfunction is frequent, occurring in approximately 35–40% of patients within 10 years of follow-up with a comparable incidence in non-ACHD as well as in ACHD patients. Any amiodarone exposure in ACHD individuals was found to be associated with an approximately 4-fold increased risk of developing thyroid dysfunction over time compared to completely amiodarone-naïve ACHD patients. Our data suggest that (i) amiodarone should be reserved for patients in whom no suitable alternatives with a more favorable side effect profile are available, (ii) patients requiring amiodarone should be evaluated prior to first prescription and continuously monitored for thyroid dysfunction and (iii) clinicians should be aware of specific risk factors associated with a higher risk of thyroid dysfunction, such as younger patient age and female gender. The suggestion that amiodarone is associated with a higher rate of thyroid dysfunction related to congenital heart disease per se is not substantiated by our data, but patients with complex ACHD may be more prone to developing complications compared to those with simple defects.

### 4.1. Incidence of Thyroid Dysfunction and Independent Risk Factors

Amiodarone is known for its antiarrhythmic properties but also for its negative effects on thyroid function [4,16,17]. Due to heterogeneity of the analyzed patient cohorts, influence of extrinsic factors such as regional iodine sufficiency on occurrence of thyroid dysfunction and differences between risk factors for amiodarone-induced thyrotoxicosis (AIT) and amiodarone-induced hypothyroidism (AIH), predictors are not uniform throughout the literature [18,19,20]. According to our nationwide dataset, ACHD patients who had been exposed to amiodarone were at increased risk for thyroid dysfunction with a hazard rate of 3.93 at four years of intake. This is consistent with the literature (irrespective of underlying diagnosis) reporting a hazard rate of 4.2 for patients enrolled in amiodarone trials on verum versus placebo therapy [21].

We identified active amiodarone intake, younger age and female gender as the main predictors for thyroid dysfunction under amiodarone intake specifically in ACHD patients. Single center data with limited patient numbers support our results: Female gender uniformly emerged as an independent risk factor for thyroid dysfunction on amiodarone [11,20]. One probable cause for female preponderance may be the excess iodine intake through amiodarone that may unmask a pre-existing subclinical thyroid disease. These are more prevalent in women due to their predisposition to autoimmune thyroid disorders [6,19].

Younger age emerged as a strong predictor for thyroid dysfunction in our analysis. While most studies agree on the association of age and thyroid dysfunction, some conflicting data exist in the literature [18,20,22,23]. For example, age was not identified as an independent predictor in the analysis by Thorne et al. [11]. This discrepancy in results may be linked to the fact that the former analysis included only ACHD patients under follow-up at a tertiary center with a narrow age distribution. In our analysis, the ACHD cohort was generally older (median age 66 years) than in the analysis by Thorne et al. (average age 35 years).

We noticed a significant association between severity of congenital heart disease and occurrence of thyroid dysfunction after amiodarone intake, but its existence per se did not emerge as a predictor. This finding is consistent with the results published by Thorne et al. [11]. Possibly, the association with complexity of disease reflects hemodynamic impairment in this patient cohort that may lead to alterations in the metabolization of drugs as well as predisposition for end-organ disease in more complex patients. Takeuchi et al. suggested that liver dysfunction in ACHD patients may be implicated with higher concentrations of circulating amiodarone, as this is not metabolized, and thus, a higher susceptibility for thyroid dysfunction [24].

Importantly, in all patients included in our analysis, chronic kidney disease, prior device implantation, smoking and alcohol abuse emerged as independent risk factors for thyroid dysfunction, emphasizing the observation that general morbidity of the patients under amiodarone may have an impact on occurrence of thyroid dysfunction.

### 4.2. Consequences of Diagnosis

Surprisingly, therapeutic consequences drawn from diagnosis of thyroid dysfunction were little. Amiodarone therapy was continued in 28% of ACHD patients with hyperthyroidism and 43% after diagnosis of hypothyroidism. Although catheter ablation may represent a curative treatment and is recommended by current guidelines as a first-choice approach for many arrhythmias, it was only performed in 42% of ACHD patients and 34% of the entire cohort in the current study [15,25].

For hypo- as well as for hyperthyroidism, termination of amiodarone intake may lead to restitution of thyroid function [26,27]. Due to the lipophilic characteristics of amiodarone, its termination will not result in an immediate elimination of the drug, as the half-life of amiodarone is not exactly known; it may be very long and seems to range between 15 and 100 days [28]. Since the benefit of discontinuation is not immediate, the prospective restitution of thyroid function must be carefully weighed against the need for antiarrhythmic drug treatment, specifically if no antiarrhythmic alternatives exist. Another aspect that must be taken into account is the higher risk for adverse outcomes of ACHD in the event of thyroid dysfunction [7]. Surprisingly, radiotherapy or thyroid surgery were infrequently performed in ACHD and in non-ACHD patients, although it is the definite treatment for hyperthyroidism in many cases [27].

### 4.3. Strengths and Limitations

Our data are based on a large set of unselected real-world data. Thus, compared to data acquired in specialized centers, the frequency distribution of patient characteristics such as complexity of congenital heart disease represents the unselected distribution of patients receiving amiodarone nationwide. The study contains no missing values, as documentation is mandatory in the German health care system for reimbursement. However, coding errors and misclassifications cannot be excluded. Although the underlying cardiac diagnosis can be identified, more detailed information about previous cardiac surgeries, such as which patient was palliated with Fontan surgery, in whom palliative surgery resulted in univentricular physiology or who was left with Eisenmenger syndrome or cyanosis, cannot be reconstructed with certainty. The number of previous cardiac catheterizations has also not been investigated due to expectable missing values, primarily because patients are insured differently when younger than 18 years. Genetic diseases have also not been investigated. As a limitation of our study, we have no data on the clinical presentation or laboratory parameters of the included patients. Thus, a detailed definition of thyroid disease is lacking, and a misclassified pseudo-hyperthyroidism via inhibition of the T4-dejodase cannot be excluded. Additionally, no statement can be made as to the frequency with which cessation of amiodarone led to normalization of thyroid function. The retrospective design and general constraints in the use of administrative care data have been described previously [29]. Furthermore, the causative context of amiodarone intake and thyroid dysfunction can only be assumed due to its time-dependent occurrence. Risk factors for thyroid dysfunction appear to be multifactorial. As they depend on regional iodine sufficiency, data should not uncritically be extrapolated to other geographic areas.

## 5. Conclusions

While amiodarone is highly effective and limited alternatives exist specifically in patients with underlying structural heart disease, the drug is associated with a high rate of thyroid dysfunction during chronic therapy. Specifically, younger patient age and female gender emerged as independent risk factors for thyroid disease. While ACHD per se was not independently associated with thyroid dysfunction, ACHD patients with more complex forms of congenital heart disease appear to be more prone to developing complications.

## Figures and Tables

**Figure 1 jcm-11-02027-f001:**
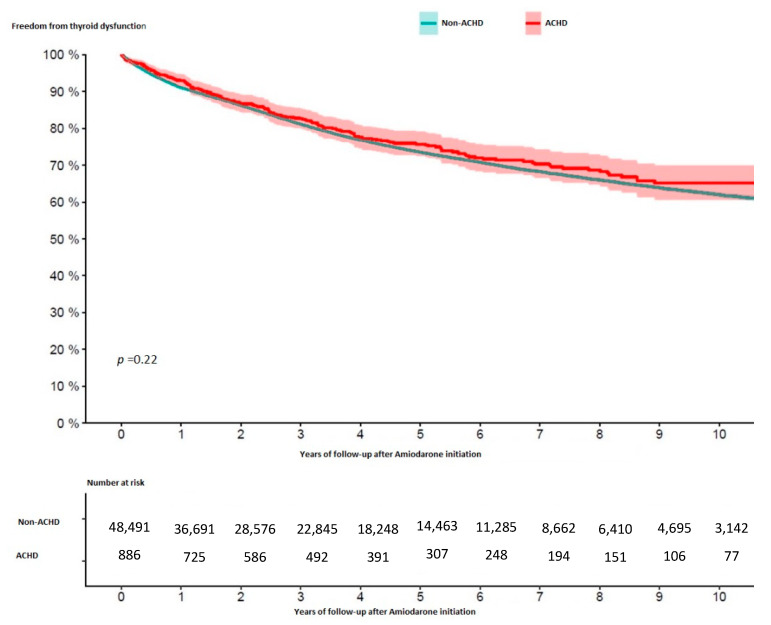
Kaplan–Meier estimates of the occurrence of thyroid dysfunction stratified between adults with congenital heart disease (ACHD) and without congenital heart disease.

**Figure 2 jcm-11-02027-f002:**
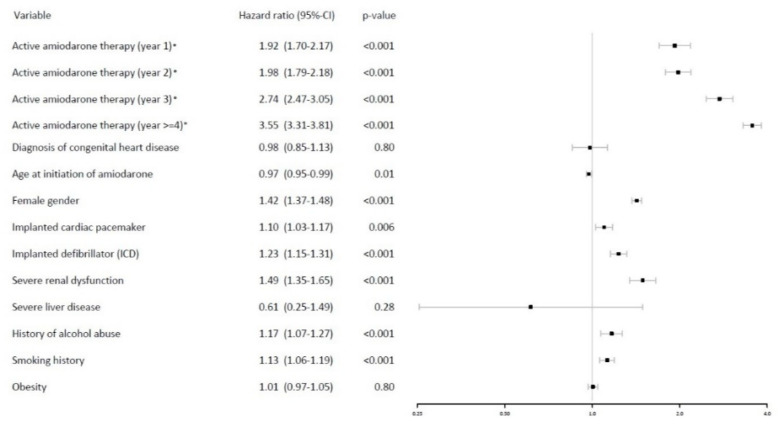
Risk factors for thyroid dysfunction based on the results of the multivariate time-dependent Cox regression analysis in all patients receiving amiodarone. * active amiodarone intake defined as prescription of amiodarone within 90 days before and after analyzation time. To avoid the problem of non-proportional hazards, a stepwise analysis showing yearly intervals was used, thus allowing the corresponding coefficient to be time-varying.

**Figure 3 jcm-11-02027-f003:**
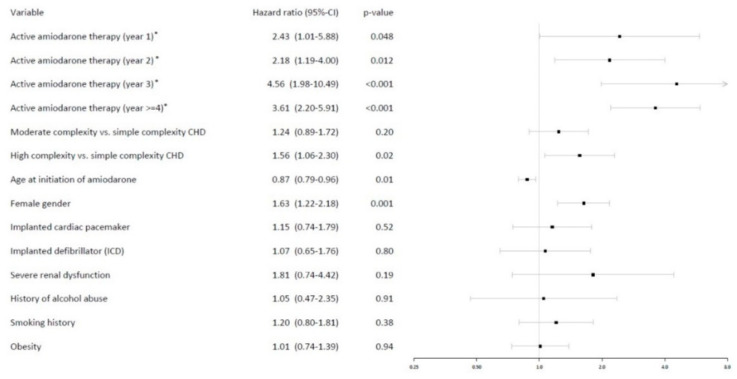
Risk factors for thyroid dysfunction based on the results of the multivariate time-dependent Cox regression analysis specifically in adult patients with congenital heart disease (ACHD) receiving amiodarone. * active amiodarone intake defined as prescription of amiodarone within 90 days before and after analyzation time. To avoid the problem of non-proportional hazards, a stepwise analysis showing yearly intervals was used, thus allowing the corresponding coefficient to be time-varying.

**Table 1 jcm-11-02027-t001:** Baseline characteristics and demographic data of the study cohort, stratified by presence and absence of congenital heart disease (ACHD = adult congenital heart disease).

	Non-ACHD, n = 48,891	ACHD, n = 886	*p*-Value
Age (median years (IQR))	73.4 (66.1–79.4)	65.9 (55.0–74.7)	**<0.001**
Sex, female, n (%)	18,066 (37.0)	301 (34.0)	0.07
Complexity of congenital heart disease			
Simple, n (%)		577 (65.1)	
Moderate, n (%)		206 (23.3)	
Severe, n (%)		103 (11.6)	
Left heart failure, n (%)	26,661 (54.5)	497 (56.1)	0.36
Right heart failure, n (%)	10,192 (20.8)	216 (24.4)	**0.01**
Pacemaker, n (%)	4118 (8.4)	81 (9.1)	0.43
Implantable cardioverter defibrillator, n (%)	4332 (8.9)	63 (7.1)	0.07
Obesity, n (%)	18.478 (37.8)	279 (31.5)	**<0.001**
Smoking, n (%)	6600 (13.5)	120 (13.5)	0.96
Alcohol abuse, n (%)	2671 (5.5)	40 (4.5)	0.23
Chronic kidney disease, n (%)	2136 (4.4)	27 (3.0)	0.06
Liver dysfunction, n (%)	53 (0.1)	0 (0.0)	1.00
Arrhythmia			
Atrial reentrant tachycardia, n (%)	7023 (14.4)	170 (19.2)	**<0.001**
Atrial fibrillation, n (%)	41,584 (85.1)	757 (85.4)	0.78
Atrial flutter, n (%)	3143 (6.4)	73 (8.2)	**0.03**
Ventricular extrasystole, n (%)	9438 (19.3)	209 (23.6)	**0.002**
Ventricular flutter/fibrillation, n (%)	2669 (5.5)	42 (4.7)	0.41
Ventricular reentrant tachycardia, n (%)	234 (0.5)	6 (0.7)	0.33
Ventricular tachycardia, n (%)	7250 (14.8)	127 (14.3)	0.74
Cardiac arrest (not specified further), n (%)	1742 (3.6)	35 (4.0)	0.52
Heart failure drug therapy			
Calcium channel blockers, n (%)	20,958 (42.9%)	331 (37.4)	**<0.001**
ACE-Inhibitors/Angiotensin II receptor blockers, n (%)	41,469 (84.8)	706 (79.7)	**<0.001**
Betablockers (excluding sotalol), n (%)	43,386 (88.7)	805 (90.9)	**0.05**
Cardiac glycosides, n (%)	11,046 (22.6)	198 (22.3)	0.90

Significant values are marked bold.

**Table 2 jcm-11-02027-t002:** Occurrence of thyroid dysfunction stratified by presence and absence of congenital heart disease (ACHD = adult congenital heart disease).

	Non-ACHD, n = 48,891	ACHD, n = 886	*p*-Value
Combined thyroid dysfunction, n (%)	10,677 (21.8)	198 (22.3)	0.71
Hyperthyroidism, n (%)	5094 (10.4)	103 (11.6)	0.24
Hypothyroidism, n (%)	7079 (14.5)	138 (15.6)	0.36

**Table 3 jcm-11-02027-t003:** Results of the propensity score-matched analysis after any amiodarone treatment in adults with congenital heart disease (ACHD) compared to amiodarone-naïve ACHD.

Variable	Hazard Ratio (95% CI)	*p*-Value
Complexity of congenital heart disease (moderate versus simple)	1.07 (0.81–1.39)	0.64
Complexity of congenital heart disease (complex versus simple)	1.48 (1.10–1.98)	0.009
Age/10 years	0.94 (0.87–1.02)	0.12
Female gender	1.78 (1.42–2.23)	<0.001
Pacemaker therapy	0.84 (0.49–1.45)	0.53
Implantable cardioverter defibrillator	0.97 (0.46–2.05)	0.94
Alcohol abuse	0.90 (0.54–1.49)	0.68
Nicotine abuse	1.26 (0.95–1.66)	0.10
Obesity	0.95 (0.75–1.22)	0.71
**Intake of amiodarone at 1 year ***	**3.68 (2.25–6.01)**	**<0.001**
**Intake of amiodarone at 2 year ***	**2.97 (1.70–5.18)**	**<0.001**
**Intake of amiodarone at 3 year ***	**4.89 (2.51–9.51)**	**<0.001**
**Intake of amiodarone at 4 year ***	**3.93 (2.52–6.12)**	**<0.001**

* Active amiodarone intake defined as prescription of amiodarone within 90 days before and after analysis time. To avoid the problem of non-proportional hazards, a stepwise analysis showing yearly intervals was used, thus allowing the corresponding coefficient to be time-varying.

**Table 4 jcm-11-02027-t004:** Overview of treatment methods after diagnosis of thyroid dysfunction stratified by presence and absence of congenital heart disease (ACHD = adult congenital heart disease).

	Non-ACHD, n = 10,677	ACHD, n = 198	*p*-Value
Levothyroxine, n (%)	4288 (40.2)	72 (36.4)	0.31
Thiamazole, n (%)	1129 (10.6)	25 (12.6)	0.35
Propylthiouracil, n (%)	36 (0.3)	3 (1.5)	**0.03**
Sodiumperchlorate, n (%)	501 (4.7)	16 (8.1)	**0.04**
Radiotherapy (thyroid), n (%)	76 (0.7)	3 (1.5)	0.17
Thyroid surgery, n (%)	170 (1.6)	4 (2.0)	0.56

Significant results are marked bold.

## Data Availability

The unblinded data included in this analysis cannot be made available due to German data protection laws (‘Bundesdatenschutzgesetz’, BDSG). They are stored on a secure drive in the Barmer health insurance database. Generally, access to data of statutory health insurance funds for research purposes is possible with respect to the German Social Law (SGB V § 287). Requests for data access can be sent as a formal proposal specifying the recipient and purpose of data transfer to the appropriate data protection agency. Access to the data can be provided to external parties under the conditions of the cooperation contract of this research project and after written approval by the health insurance fund.

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
