# Peer review of "Thyroid Dysfunction under Amiodarone in Patients with and without Congenital Heart Disease: Results of a Nationwide Analysis"

_jcm, 2022, doi:10.3390/jcm11072027_

Round 1

Reviewer 1 Report

With the present research, Dr Fischer and colleagues aimed to describe the incidence and risk factors of amiodarone-induced thyroid dysfunction in adults with congenital heart disease (ACHD) compared to the general population. This is an attractive topic as, thyroid disease are currently emerging as a novel risk factor for adverse events in patients with acquired heart disease. I sincerely congratulate the authors for their choice of investigating amiodarone-induced thyroid dysfunction, which is a particularly worrisome event in the complex ACHD population.

Nevertheless, I would like to point out some limitations and suggest several changes, which could improve the manuscript quality:

  • Firstly, a detailed definition of thyroid dysfunction is lacking throughout the manuscript. The authors should clarify whether a cut off value of thyroid hormones was used for the definition of hypo-hyperthyroidism and whether subclinical thyroid dysfunction was also included.
  • Secondly, additional demographical and clinical data are required for both the general population and ACHD group. In particular for the ACHD group, I strongly believe that it is mandatory to report details on the cardiac diagnosis, number of cardiac surgeries/catheterization, genetic disease, proportion of patients with cyanosis, univentricular physiology and/or Fontan palliation, and Eisenmenger syndrome, as it has been extensively described that those factors may influence the thyroid function.
  • The authors should describe in detail how was the amiodarone-naïve ACHD population enrolled, which variables and which caliper were used in the propensity matching score, and how the matching was performed (1:1, 1:2, etc.).

Other minor limits are:

  • In the methods section, the authors stated that “following a 10-day initiation period after first prescription of amiodarone, patients were screened for documented thyroid dysfunction (defined as hypo-, hyperthyroidism, thyroiditis and/or thyroid surgery/radiotherapy) as well as occurrence of hypothyroidism or hyperthyroidism only”. This sentence is unclear. Moreover, the authors should further explain how was a thyroid function screening performed 10 days after treatment initiation in a retrospective study design.
  • As disease complexity configures a condition of increased risk of developing thyroid dysfunction, the incidence of amiodarone-induced thyroid disease in the ACHD group might have been influenced by the large prevalence of simple defects in the study population. This issue should be discussed and acknowledged in the limitations of the study.
  • How were severe renal and liver disease defined and why was severe liver disease not included in the multivariable analyses for the ACHD group?
  • Can the authors specify whether amiodarone cessation rate was significantly higher in the ACHD group? If so, how do the authors comment on this finding?
  • What was the thyroid function normalization rate after amiodarone cessation and was it different in the 2 groups?

  • As the prescription of amiodarone should be carefully weighed against potential risks, it would be interesting if the author could also extrapolate data on treatment efficacy on arrhythmia control in ACHD.
  • The ACHD amiodarone-naïve population and propensity matching score as well as the important finding of higher risk of amiodarone-induced thyroid dysfunction in those with complex defects were not mentioned in the abstract.
  • Table 3 legend is unclear and should be clarified.
  • Multivariable Cox analysis and propensity matching score methodology should be described in the statistical analysis paragraph and not at beginning of the methods section.
  • Finally, even though I am well aware of the journal policy of not allowing the reviewers to suggest their own works as reference, I would like invite the editors and the authors to consider the following citation:

“Fusco F, Scognamiglio G, Guarguagli S, Merola A, Palma M, Barracano R, Borrelli N, Correra A, Grimaldi N, Colonna D, Roma AS, Romeo E, Sarubbi B. Prognostic Relevance of Thyroid Disorders in Adults With Congenital Heart Disease. Am J Cardiol 2021:S0002-9149(21)01142-5.”

To the best of my knowledge, this is the only report of increased risk of negative outcome in ACHD patients with thyroid disorders. This previous finding may further contribute to raise the scientific value of the present research, providing a solid base to explain the importance of researching factors implicated in the development of amiodarone-induced thyroid dysfunction in this vulnerable population.

Reviewer 2 Report

Dear authors, than you for the opportunity of rewieving "Thyroid Dysfunction under Amiodarone in Patients with and 2 without Congenital Heart Disease: Results of a Nationwide Analysis" by Fisher et al.

Please, find my comments below:

  1. INTRODUCTION : sometimes, Amiodaron is used in CHD patients because other drugs such as beta-blockers or sotalol fail to treat the arrhythmia; please discuss this.
  2. DISCUSSION: it has been demonstrated that thyroid diseases are predictive of adverse outcome in the ACHD population (  Prognostic Relevance of Thyroid Disorders in Adults With Congenital Heart Disease. Fusco F, Scognamiglio G, Guarguagli S, Merola A, Palma M, Barracano R, Borrelli N, Correra A, Grimaldi N, Colonna D, Roma AS, Romeo E, Sarubbi B.Am J Cardiol. 2022 Mar 1;166:107-113). Please, discuss this
